# The Relationship between Meaning in Life and Depression among Chinese Junior High School Students: The Mediating and Moderating Effects of Cognitive Failures and Mindfulness

**DOI:** 10.3390/ijerph20043041

**Published:** 2023-02-09

**Authors:** Ying Li, Yihan Jin, Huiyan Kong, Chao Feng, Lei Cao, Tiantian Li, Yue Wang

**Affiliations:** School of Education, Zhengzhou University, Zhengzhou 450001, China

**Keywords:** meaning in life, depression, cognitive failures, mindfulness, junior high school students, adolescents

## Abstract

In recent years, the incidence of depression among adolescents has been increasing yearly, and the severe damage of depression on adolescents’ physical and mental health development has caused extensive concern worldwide. Previous research on adults has confirmed that meaning in life is a crucial buffer factor for depression, and developing meaning in life is an essential task in adolescence. Moreover, prior researchers also pointed out that frequent cognitive failures can induce negative emotions in individuals, whereas mindfulness can regulate individuals’ depression levels. However, few studies have investigated the impact of meaning in life on depression in adolescents and the underlying psychological mechanisms. Accordingly, based on the theoretical framework of the Cognitive Vulnerability–Stress Theory of Depression, this study aimed to explore the relationship between meaning in life and depression in junior high school students, as well as the mediating effect of cognitive failures and the moderating effect of mindfulness. We collected data from 948 adolescents aged 11 to 17 in two junior high schools in Henan Province, China, and tested the theoretical model through the PROCESS macro for SPSS. The results showed that: (1) meaning in life had a significant negative predictive effect on depression (*β* = −0.24, *p* < 0.001); (2) cognitive failures partially mediated the relationship between meaning in life and depression (*β* = 0.31, *p* < 0.001); (3) the relationship between cognitive failures and depression was moderated by mindfulness (*β* = −0.05, *p* < 0.05). This study implied that we could start by cultivating adolescents’ meaning in life and improving their level of mindfulness to prevent and intervene in adolescent depression.

## 1. Introduction

Due to the imbalance of physical and psychological development, children in adolescence are more likely to be affected by external and internal factors that cause various mental health problems. One of the mental illnesses that adolescents frequently suffer from is adolescent depression. Adolescence is the peak period of depression. A systematic review in 2022 pointed out that the incidence rate of depression among adolescents in the world is as high as 34% [1]. The detection rate of depressive symptoms among Chinese junior high school students ranged from 6.4% to 60.7%, with a combined value of 28.4%, according to a recent meta-analysis study [2]. Depression has a negative impact on adolescents’ regular learning and life, as well as their physical and mental development, resulting in a drop in academic performance, poor interpersonal relationships, anxiety, insomnia, a sense of worthlessness in life, and even suicidal behavior [3,4]. As the global incidence of depression among adolescents increases yearly [5], it is urgent and necessary to explore the influencing factors and potential mechanisms of depression in junior high school students.

### 1.1. Meaning in Life and Depression in Junior High School Students

Adolescence is the second leap in the development of self-awareness. The inner world of adolescents becomes richer, and they begin to spend much time reflecting on the meaning of life. Meaning in life was derived from Frankl’s logotherapy theory, which refers to people’s understanding and pursuit of the purpose and goal of life. Frankl proposed that obtaining and maintaining meaning in life is one of the primary motives of human beings [6]. Steger et al. have divided meaning in life into two dimensions, the perception dimension of “the presence of meaning in life” and the motivation dimension of “the search for meaning in life” [7]. As an essential life experience, meaning in life is a crucial factor in maintaining individual mental health [8]. In general, individuals with a higher meaning in life also have higher levels of mental health [9]. In contrast, individuals who lack meaning in life are more likely to experience anxiety, depression, and other negative emotions and even appear to have suicidal ideation and suicidal behavior in severe cases [10,11]. Meaning in life is also an essential buffer against depression [12,13]. Studies in adults and college students have shown that meaning in life can reduce depression caused by adverse events and significantly negatively predict depression [14,15]. Clinical studies also stated that meaning in life has a positive therapeutic effect on depressed individuals [16]. Based on these findings, we can infer that meaning in life is closely related to depression. 

Most of the existing research on the connection between depression and meaning in life focuses on adults, particularly college students. However, there are few studies on the group of junior high school students who are in adolescence. Adolescence is a crucial period for developing meaning in life [17], and adolescents generally have low levels of meaning in life [18], which may be one of the leading causes of the high incidence of depressive episodes during this period. Clarifying the relationship between meaning in life and depression in junior high school students can help teenagers better understand themselves, improve their mental health, and provide new research insight for the prevention of adolescent depression. Therefore, the current study aimed to illuminate the relationship between meaning in life and depression in junior high school students and the underlying mechanisms. Accordingly, we propose Hypothesis 1:

**Hypothesis 1.** *Meaning in life negatively predicts depression in junior high school students*.

### 1.2. Cognitive Failures as a Mediator

Cognitive failures are cognition-based slips and lapses in simple tasks that a person can usually perform without errors, including attention, memory, and motor failures [19]. For example, entering a room but forgetting what to look for, throwing away something that should be kept, and so on. Some studies have pointed out that the occurrence of cognitive failures may be related to insufficient cognitive resources [20], negative affects [21], and personal traits such as trait anxiety and neuroticism [22,23]. Cognitive failures can easily cause negative emotions in individuals, affect their work and learning efficiency, and damage their mental health [24]. It has been established that there is a significant correlation between cognitive failures and depression, and individuals who experience more cognitive failures have a higher level of depression [25,26]. Furthermore, repeated cognitive failures tend to cause negative self-evaluations [27], which can increase the risk of depression [28]. Conversely, the cognitive functions of depressed individuals, including processing speed, logical reasoning, attention, and executive function, will decline to varying degrees, which are also essential factors in triggering cognitive failures [29]. Thus, it may form a vicious cycle between depression and cognitive failures.

Although no empirical studies have directly demonstrated the intrinsic relationship between meaning in life and cognitive failures, relevant theories and research findings have implied that meaning in life is closely related to cognitive failures. Resource Conservation Theory [30,31] believes that the accumulation of individual negative emotions will consume cognitive resources. Meaning in life is essentially a subjective experience, inseparable from emotional states [32], and might thereby also be a consumer of cognitive resources. Moreover, individuals lacking meaning in life tend to have negative self-cognition and self-evaluation, which is the driving force that increases cognitive failures [33]. Furthermore, individuals with low meaning in life have a poor ability to concentrate, and attention deficit is the basis of cognitive failures. When there are external distractions, their attention easily departs from the current task they are engaged in, resulting in attention failures [21]. It has been also noted that meaning in life is significantly associated with boredom, which is one of the valid predictors of cognitive failures [29]. Reversely, frequent cognitive failures will also cause individuals to worry that they will continue to make mistakes and thus experience enormous mental stress [34]. As a crucial stress-coping resource for individuals [35], meaning in life can mitigate the adverse effects of stress and, thus, reduce cognitive failures. 

Based on the above analysis and summary, this study proposes Hypothesis 2: 

**Hypothesis 2.** *Meaning in life negatively predicts the depression of junior high school students through the mediating effect of cognitive failures*.

### 1.3. Mindfulness as a Moderator

Mindfulness refers to the state of consciousness attained by people when they pay intentional, non-judgmental attention to the situation at hand, with particular emphasis on attention and awareness of the present [36]. According to the Mindfulness Stress Buffering Hypothesis, mindfulness practice can lessen the unfavorable effects of stress and adversity on mental health [37]. Empirical studies found that mindfulness has positive impacts on reducing negative emotions and improving well-being [38,39]. Additionally, clinical research has demonstrated that mindfulness training can considerably alleviate depressive symptoms in people with depression, postpartum depression, and recurrent depressive disorder [40,41] and improve their physical symptoms and sleep [42,43,44]. Furthermore, neurophysiological evidence also suggested that individuals with high levels of mindfulness have reduced activity in the amygdala and activation in the left prefrontal region, indicating an increase in positive emotions and a decrease in negative emotions [45]. The above suggests that mindfulness level is strongly correlated with depression.

The mindfulness level also affects individuals’ cognitive function. The cognitive model of mindfulness points out that mindfulness intervention can enhance the individual’s selective perception and working memory, thus inhibiting interfering information and improving the individual’s executive function [46], and individuals with high levels of mindfulness have stronger cognitive flexibility [47]. Empirical studies also pointed out that mindfulness is an essential endogenous factor of cognitive failures [48], and mindfulness training can significantly reduce the incidence of cognitive failures [49]. 

Regarding the relationship between mindfulness and meaning in life, a meta-analysis has indicated a significant positive correlation between mindfulness and meaning in life in college students. Mindfulness can positively predict meaning in life by enhancing self-awareness [50]. According to the self-determination theory, individuals can perceive meaning in life when they live in a way that is consistent with the self. It can be inferred that individuals with a high level of mindfulness are more able to express themselves honestly, dominate their behavioral performance, and, thus, have a higher meaning in life. In short, a high level of mindfulness is a protective factor in alleviating individual negative cognition and emotions. Based on the above theoretical analysis and empirical research, we propose Hypothesis 3:

**Hypothesis 3.** *Mindfulness plays a moderating role in the mediating model of meaning in life, cognitive failures, and depression*.

### 1.4. The Present Study

In summary, most existing studies on the relationship between meaning in life and depression focus on college students or other adult groups. Little is known about junior high school students who are still in the critical period of self-identity development. Accordingly, this study aims to evaluate the intrinsic association between meaning in life and depression in junior high school students, as well as the potential mechanism of the impact of meaning in life on depression, to provide a new research direction for the prevention and intervention of depression in adolescents.

The Cognitive Vulnerability–Stress Theory of Depression [51] illustrates that cognitive vulnerability and negative events interact with each other to influence the occurrence of depression. Frequent cognitive failures, as negative events, can evoke more negative emotions in individuals. Meaning in life and mindfulness can also be considered as individual cognitive vulnerabilities. Therefore, this study hypothesized that the interaction of cognitive failures, meaning in life, and the level of mindfulness could jointly affect the generation and development of depression. Based on the above research analysis and theoretical framework, we constructed a moderated mediation model (as shown in Figure 1). Three main questions were explored in current study: (1) the effect of junior high school students’ meaning in life on depression; (2) the mediating effect of cognitive failures between meaning in life and depression; and (3) whether the level of mindfulness moderates each pathway.

## 2. Methods

### 2.1. Participants

In this study, we recruited 22 classes of junior high school students as participants from two junior high schools in Henan Province, China. A total of 1100 questionnaires were distributed, 152 invalid questionnaires (including incomplete and regular questionnaires) were excluded, and 948 (86.2%) valid questionnaires were obtained. The average age of all participants was 13.70 years old (*SD* = 1.22, range from 11 to 17), including 436 males (46%) and 512 females (54%). They came from three grades, including 387 students (40.8%) in Grade One, 288 students (30.4%) in Grade Two, and 273 students (28.8%) in Grade Three. There were 22 only-children and 926 non-only-children. 

### 2.2. Procedures

After being approved by the Ethical Committee for Scientific Research at the authors’ institution and obtaining the informed consent of the school, teachers, and students, we conducted this survey using a whole-group random sampling method in a classroom setting and assigned trained psychology graduate students as experimenters. Students who participated in this survey were asked to complete the questionnaires independently according to their true feelings. All questionnaires were collected anonymously and retrieved on the spot after the test was completed.

### 2.3. Measures

#### 2.3.1. Meaning in Life

This study adopts the Meaning in Life Questionnaire, compiled by Steger et al. and revised by Liu and Gan [52]. The scale contains nine items divided into two dimensions: the presence of meaning in life and the search for meaning in life. The scale is rated on a 7-point Likert scale, with one representing “totally disagree” and seven representing “totally agree”. The sum of the scores of all items is the total score, and the higher the score, the higher the individual’s meaning in life. The scale has been widely used in China [53]. The reliability and validity of the scale have been validated in adolescents [54]. The Cronbach’s alpha coefficient of the scale in this study is 0.74.

#### 2.3.2. Depression

The study used a 13-item version of the BDI compiled by American psychologist Beck AT (1974), revised by Zheng et al. [55], tested with good reliability and validity, and has been widely used in Chinese adolescents [27,56]. Each item was scored with a 4-point score ranging from 0 to 3 (no such symptoms = 0; mild = 1; moderate = 2; severe = 3), with higher scores indicating higher depressive proneness and lower scores indicating lower depressive proneness. A score of 0 to 4 represents no depressive symptoms; 5 to 7, mild; 8 to 15, moderate; and 16 or more, severe. The Cronbach’s alpha coefficient for this scale in this study was 0.90, with good internal consistency.

#### 2.3.3. Cognitive Failures

This study used the Chinese version of the Cognitive Failures Questionnaire compiled by Broadbent et al. (1982) [34] and revised by Zhou et al. [57]. The scale has good reliability and validity and has been widely used in Chinese adolescents [58,59]. The scale has 25 items and is scored on a 5-point scale (from “1 = never” to “5 = always”) and includes five dimensions: interference, memory, interpersonal error, motor coordination, and name memory, with higher scores representing higher levels of cognitive failures. The Cronbach’s alpha coefficient for this scale in this study was 0.92.

#### 2.3.4. Mindfulness

The Chinese version of the Mindful Attention Awareness Scale was compiled by Brown and Ryan (2003) and revised by Chen et al. [60]; the scale has good reliability and validity and has been widely used in Chinese adolescents [61,62,63]. The 15-item scale uses a 6-point scale ranging from 1 “always” to 6 “never,” and there are reverse rating questions. A higher total score indicates a higher level of mindfulness among adolescents. The Cronbach’s alpha coefficient of the scale in this study was 0.86.

### 2.4. Data Analysis

In this study, SPSS 22.0 was used for factor analysis and common method bias test. First, descriptive statistical analysis was made on the demographic variables of valid questionnaires. Second, Pearson correlation analysis was used to calculate the correlation coefficients between the variables. Finally, model 59 and model 14 in Hayes’ PROCESS macro for SPSS were used to test the mediating and moderating effects.

## 3. Results

### 3.1. Common Method Bias Test

Harman’s One-factor Test was used to test for common method bias. The unrotated principal component and factor analysis for all items showed that there were 11 factors with eigenvalues greater than 1, and the factor with the highest explanatory variation rate was 22.70%, far below the threshold of 40%, indicating that there was no common method bias.

### 3.2. Descriptive Statistics and Correlation Analysis

Table 1 lists each studied variable’s means, standard deviations, and correlation matrices. The results showed that depression had significant negative correlations with meaning in life and mindfulness and a significant positive correlation with cognitive failures. There was a significant negative correlation between meaning in life and cognitive failures and a significant positive correlation between meaning in life and mindfulness. Cognitive failures were significantly negatively correlated with mindfulness. In addition, gender and age significantly correlated with some variables, so were treated as control variables in the subsequent analysis.

### 3.3. Testing for Moderated Mediation Effect

A moderated mediation model test was conducted according to the suggestions of Wen and Ye [64]. After standardizing all continuous variables, model 59 in the PROCESS macro program was used to test whether mindfulness played a moderating role in the mediation model. Model 14 was further applied to examine whether there was a moderating role of mindfulness in the relationship between cognitive failures and depression. 

The results showed that meaning in life has a significant negative predictive effect on depression (*β* = −0.24, *p* < 0.001), which confirmed Hypothesis 1. When the mediating and moderating variables were included, the direct predictive effect of meaning in life on depression remained significant (*β* = −0.17, *p* < 0.001); moreover, the negative predictive effect of meaning in life on cognitive failures was significant (*β* = −0.14, *p* < 0.001), and cognitive failures had a positive predictive effect on depression (*β* = 0.31, *p* < 0.001), which indicated that cognitive failures partially mediated the relationship between meaning in life and depression, so Hypothesis 2 holds. Furthermore, the interaction term of cognitive failures and mindfulness had a significant predictive effect on depression (*β* = −0.05, *p* < 0.05), suggesting that mindfulness played a moderating role in the effect of cognitive failures on depression, i.e., mindfulness moderated the second half of the mediation model’s pathway, and Hypothesis 3 partially holds. Details are presented in Table 2.

To clarify the moderating effect trend of mindfulness, we divided mindfulness into the low and high groups (1 *SD* below the mean and 1 *SD* above the mean). We used a simple slope test to investigate the effect of cognitive failures on depression at different levels of mindfulness. As shown in Figure 2, when the level of mindfulness was high, cognitive failures had a weak positive predictive effect on depression (*β* = 0.25, *t* = 5.44, *p* < 0.001); when the level of mindfulness was low, the positive predictive effect of cognitive failures on depression was significantly enhanced (*β* = 0.36, *t* = 8.83, *p* < 0.001); that is to say, as the level of mindfulness increased, the positive predictive effect of cognitive failures on depression diminished.

## 4. Discussion

Previous studies on adults have indicated that meaning in life has a negative predictive effect on depression, but there are few studies on adolescents. By introducing cognitive failures and mindfulness as mediating and moderating variables, the current study further investigated the underlying psychological mechanism of the relationship between meaning in life and depression in junior high school students. Moreover, we try to reveal “how meaning in life works” and “when mindfulness plays a better role.” The results found that: (1) meaning in life negatively predicted depression in junior high school students; (2) cognitive failures played a partial mediating role in the relationship between meaning in life and depression; and (3) the level of mindfulness moderated the relationship between cognitive failures and depression; that is to say, the higher the level of mindfulness, the weaker the positive predictive effect of cognitive failures on depression.

### 4.1. The Relationship between Meaning in Life and Depression

As shown by the results, meaning in life in junior high school students significantly negatively predicted depression, which supported Hypothesis 1 and was consistent with the previous research finding [65]; that is, the higher the individual’s meaning in life, the lower the level of depression. This negative correlation has also been confirmed in the adult group, which indicates that meaning in life, as an essential buffer factor for depression, has cross-age consistency. According to Erikson’s theory of psychosocial development, the primary developmental task of adolescents is to establish self-identity, which coexists and positively correlates with meaning in life [66]. Adolescents begin to think about “who I am” and “what kind of person I want to be”. They need to interact with the external environment to improve and integrate themselves and understand the meaning of life. However, junior high school students are in a special period of physical and mental development. Adolescents in this period are emotionally unstable, rebellious, and have weak psychological endurance. When confronting negative factors such as increased growth confusion, changes in the school environment, increased academic pressure, and being unable to cope with these problems successfully, adolescents tend to evaluate themselves negatively, which can lead to depression [67]. According to dynamic characteristics, junior high school students with low meaning in life have difficulties identifying their goals and missions, and view their surroundings as monotonous, and tend to connect unpleasant experiences with themselves, which makes them feel helpless and incompetent, and, thus, they are more likely to develop depression. In contrast, from the perspective of positive psychology, clear life goals can mitigate the adverse effects of various conflicts experienced by adolescents and help them deal with difficulties and setbacks with a positive attitude. Problem-solving can also improve their self-efficacy and self-confidence, thereby reducing the risk of depression [68].

### 4.2. The Mediating Role of Cognitive Failures

The current study found that cognitive failures mediated the relationship between meaning in life and depression in junior high school students. We can explain the result from the perspective of “motivation → cognition → emotion”. Junior high school students who lack meaning in life tend to have weak learning motivation [69]. They are more likely to suffer from inattention and wandering thinking, leading to cognitive failures, and further induce anxiety and depression.

Junior high school students undergo the “stormy” and “psychologically weaned” phases. Although they have high levels of independence and self-consciousness, they have poor emotion control skills, and the accumulation of unpleasant feelings will drain their psychological resources. When cognitive resources are lacking, sustained attention will fail [70]. Moreover, adolescents with low meaning in life are confused and anxious about their future, so they are more likely to be distracted and waste time and energy on boring and useless things, and then they may even make more mistakes when completing simple tasks. Therefore, junior high school students who lack meaning in life may experience more cognitive failures.

Additionally, junior high students’ dialectical thinking is still immature, and they tend to view problems in a one-sided and isolated way. When negative events (such as frequent cognitive failures) occur, they are more prone to self-denial and have pessimistic thoughts. Long-term self-denial and negative cognition tend to increase the possibility of depression.

### 4.3. The Moderating Role of Mindfulness

The risk-protective factor model argues that protective factors (such as mindfulness level) and risk factors (such as cognitive failures) work together in adolescent development [71]. Risk factors may drive individuals in an undesirable direction, whereas protective factors can buffer this trend. The current study found that the level of mindfulness moderated the relationship between cognitive failures and depression in junior high school students; that is, mindfulness weakened the positive predictive effect of cognitive failures on depression, indicating that mindfulness is an essential protective factor for mental health and personality improvement. Junior high school students with high levels of mindfulness tend to be less depressed, possibly because even when they are disengaged from the current task due to inattention, mindfulness helps them refocus on the present moment. Therefore, mindfulness can reduce the adverse effects of anxiety, low self-esteem, and lack of confidence that may result from cognitive failures, and further reduce the level of depression.

We further found that cognitive failures had a significant positive predictive effect on depression for students with low levels of mindfulness. In contrast, for junior high school students with high levels of mindfulness, the positive predictive effect of cognitive failures on depression was weaker. In other words, the positive predictive effect of cognitive failures on depression in junior high school students gradually diminished as mindfulness increased. This finding can be explained from the following perspectives. First, mindfulness focuses on the present moment in a non-judgmental and accepting way, which can reduce redundant thoughts about negative emotions such as depression. Students with high mindfulness levels have better insight into their inner world. They can timely and consciously adjust their attention content, so they will only be immersed in the harmful effects of cognitive failures for a short time. Second, according to the Broadened-build Theory of Positive Emotions [72], mindfulness, as a positive psychological variable, can expand the scope of attention, enhance cognitive function, and increase positive psychological resources such as mental toughness, self-efficacy, and self-esteem [42,73]. Moreover, this result was also in line with the Social Emotional Selection Theory [74]. Students with high levels of mindfulness have more sensitive time perception and pay more attention to the present experience and positive information. This assists individuals in better coping with the negative consequences of cognitive failures, reducing the likelihood of depressive moods. 

We also examined the moderating effect of mindfulness on the relationship between meaning in life and cognitive failures, as well as meaning in life and depression, respectively. We found that the moderating effect in these two paths was insignificant, indicating that the impact of meaning in life on cognitive failures or depression in junior high school students did not vary with the level of individual mindfulness. This result might be because mindfulness has a limited effect on protecting against high-risk factors [75], which is also consistent with the “drop in the bucket” regulation model [76]; that is, an increase in the level of mindfulness is more beneficial for individuals in low-risk contexts. On one hand, a low level of meaning in life is a high-risk factor for junior high school students whose self-identity is not fully formed, and its influence is more direct and robust. On the other hand, personal values and beliefs are difficult to change once established, and discovering the purpose of life is a lengthy process. Therefore, the level of mindfulness cannot moderate the strong influence of meaning in life. This result also suggested that we should be more cautious in drawing research conclusions that mindfulness has a positive effect on individual mental health. The mechanism and scope of the effect of mindfulness should be carefully examined rather than being overstated.

### 4.4. Implications and Limitations

This study revealed the intrinsic mechanism of the relationship between meaning in life and depression in junior high school students, which has significant theoretical value and enlightening significance for reducing the occurrence of depression in junior high school students and strengthening the prevention and intervention of depression. First, meaning in life negatively predicts depression in junior high school students. Parents and schools should attach importance to cultivating teenagers’ meaning in life, set up life education courses, and guide students to think about the meaning of life and improve their meaning in life. Second, cognitive failures are closely related to insufficient cognitive resources. Therefore, setting a clear life goal and maintaining an optimistic emotional state can reduce the depletion of cognitive resources by negative emotions such as boredom and anxiety, thus reducing the occurrence of cognitive failures and depression. Finally, mindfulness plays a prominent regulatory role in depression, which suggests that we should take specific measures to improve adolescents’ mindfulness levels. Furthermore, schools can conduct mindfulness lectures and group counseling to improve adolescents’ mindfulness levels, reduce their excessive attention to conflict and frustration, improve their cognitive and behavioral flexibility, and, thus, reduce the incidence of depression.

Indeed, this study has some limitations that need to be improved in future studies. First, this study used a cross-sectional research design, which could not reveal the causal relationships between variables. Future studies can use experimental or follow-up research to explore the causal relationship and long-term effects between variables. Second, the Mindful Attention Awareness Scale used in this study is a single-dimension questionnaire, which lacks the measurement of different dimensions of mindfulness and, therefore, cannot explore which components of mindfulness are more effective in moderation. Third, the data collected in this study came from subjective reports of adolescents and may be subject to some error (e.g., memory bias, social desirability). Future studies could consider collecting data from other sources, such as teachers, parents, and peers, to measure relevant variables objectively. Fourth, this study used traditional linear regression analysis to process the data, which failed to consider the impact of different subject groups on the research results. In subsequent investigations, using more comprehensive data analysis methods, such as HLM models, to analyze the data is recommended. Finally, the participants in this study come from the provincial capital of central China, so the findings obtained in this study can be extended to urban junior high school students. Because of the educational quality gap between urban and rural areas in China, it is still being determined whether the results can be generalized to rural junior high school students. Future studies can compare the differences in depression status between rural and urban junior high school students and analyze the related influencing factors.

## 5. Conclusions

Although previous studies have explored the relationship between meaning in life and depression in adults, there are few studies on adolescents. This study examined the relationship between meaning in life and depression in junior high school students and its underlying psychological mechanisms. Our findings suggested that meaning in life in junior high school students can negatively and significantly predict depression or indirectly predict depression through cognitive failures, and mindfulness levels can moderate the relationship between cognitive failures and depression.

## Figures and Tables

**Figure 1 ijerph-20-03041-f001:**
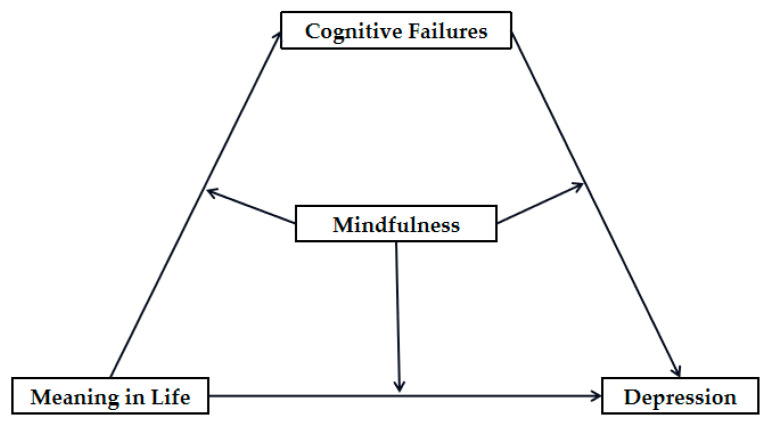
The proposed moderated mediation model.

**Figure 2 ijerph-20-03041-f002:**
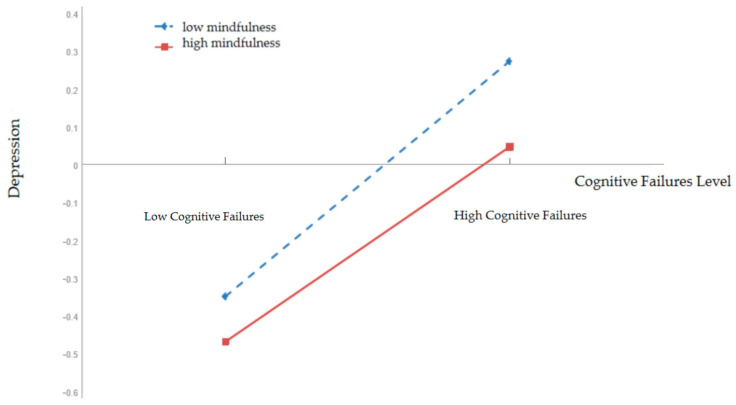
Mindfulness as a moderator of cognitive failures and depression (*n* = 948).

**Table 1 ijerph-20-03041-t001:** Descriptive statistics and correlations among variables (*n* = 948).

	*M*	*SD*	1	2	3	4	5	6
1 Age	13.70	1.22	1	-	-	-	-	-
2 Gender	-	-	-	1	-	-	-	-
3 Meaning in life	38.78	8.06	−0.05	−0.09 **	1	-	-	-
4 Cognitive failures	68.51	16.51	0.14 **	0.07 *	−0.1 5 **	1	-	-
5 Mindfulness	54.09	12.70	−0.18 **	−0.00	0.14 **	−0.64 **	1	-
6 Depression	10.27	7.25	0.19 **	0.05	−0.25 **	0.46 **	−0.41 **	1

*M*, mean; *SD*, standard deviation; * *p* < 0.05, ** *p* < 0.01.

**Table 2 ijerph-20-03041-t002:** Testing the moderated mediation model (*n* = 948).

	Depression			Cognitive Failures			Depression		
	*β*	*t*	95% *CI*	*β*	*t*	95% *CI*	*β*	*t*	95% *CI*
Gender	0.04	1.09	[−0.03, 0.10]	0.06	1.91	[−0.00, 0.12]	0.01	0.36	[−0.05, 0.06]
Age	0.17	5.66 ***	[0.11, 0.24]	0.13	4.29 ***	[0.07, 0.19]	0.11	3.85 ***	[0.05, 0.16]
Meaning in life	−0.24	−7.73 ***	[−0.30, −0.18]	−0.14	−4.42 ***	[−0.20, −0.08]	−0.17	−6.04 ***	[−0.22, −0.11]
Cognitive failures							0.31	8.25 ***	[0.24, 0.39]
Mindfulness							−0.17	−4.66 ***	[−0.25, −0.10]
Cognitive failures × Mindfulness							−0.05	−2.26 *	[−0.10, −0.01]
*R^2^*	0.10			0.05			0.28		
*F*	33.00 ***			14.94 ***			60.65 ***		

* *p* < 0.05, *** *p* < 0.001.

## Data Availability

The data presented in this study are available in article.

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
