# Peer review of "The Relationship between Meaning in Life and Depression among Chinese Junior High School Students: The Mediating and Moderating Effects of Cognitive Failures and Mindfulness"

_ijerph, 2023, doi:10.3390/ijerph20043041_

Round 1
Reviewer 1 Report
Interesting, well-written paper. Nevertheless some issues need to be adressed
Abstract: in is unclear what exactly the Authors mean by "cognitive failure".
Intruduction: another important construct explored in the study is "meaning in life". Althought the idea may be intuitively understood I would appreciated more detailed referring to the theoretical background of the "meaning in life" idea. Do Authors acknowledge "meaning in life" as psychological construct, transcendent idea, others?
Similarily, although the Authors descibe the meaning of "cognitive failures" there is no explanation provided how they interpret this phenomenon (cognitive impairment? neurodevelopmental deficits? reactive disturbances?)
Although the Authors discuss some mechanisms which may be resposible for association betwieen meaning, in life and cognitive failures and mindfullness I would appreciate some more consistent theoretical framework linking these ideas.
Method
The Authors mentioned that the vast majority of the participants were not-only children. Forgive my ignorance: taking into consideration the one-child policy in China I wonder how it is possible. Could you explain that to me and other non-Chinese readers?
Psychometric tools: it should be clarified if the questionnaires were validated for children and adolescents age group
Discussion
In the results desription and in the disussion section the Authors mention that "meaning in life affects depression". This and other similar expressions suggest the causative relationship between meaning in life and depression. Similar interpratations are suggested for mindfullness and cognitive failures. Such conclusion definitely cannot be drawn from a cross-sectional study. Corss-sectional studies can detect an association, but not causative effects. Maybe lack of meaning in life is just another depressive symptom, not the cause of depression?
Implications
In the implications section the Authors present opinins which are not supperted by the study results (e.g. "Schools and families should guide teenagers to use mobile phones properly to avoid them wasting too much time and energy on mobile phones" - the study is not about mobile phones!) This should be avoided and the section rewritten.
Author Response
Point 1: Abstract: in is unclear what exactly the Authors mean by "cognitive failures".
Response 1: Thank you for your valuable comments. To clarify the research variables of this study more clearly, we have supplemented the abstract with a brief introduction to cognitive failures and mindfulness, adding the following----”Cognitive failures refer to small lapses in attention, memory, or movement that everyone experiences in daily life, and has been confirmed by empirical studies to be closely related to depression. Mindfulness is the non-judgmental awareness of the present physical and mental experience. Individuals with high levels of mindfulness tend to have less negative emotions.”
Point 2: Introduction: another important construct explored in the study is "meaning in life". Although the idea may be intuitively understood I would appreciated more detailed referring to the theoretical background of the "meaning in life" idea. Do Authors acknowledge "meaning in life" as psychological construct, transcendent idea, others?
Response 2: Thank you for your question. We added the theoretical background and dimensions of meaning of life in the manuscript as follows “Meaning in life was derived from Frankl's logotherapy theory, which refers to people's understanding and pursuit of the purpose and goal in their life. Frankl proposed that obtaining and maintaining meaning in life is one of the primary motives of human beings[6]. Steger et al. have divided meaning in life into two dimensions, the perception dimension of "the presence of meaning in life" and the motivation dimension of "the search for meaning in life"[7]”, hoping to clarify the concept better. As far as we understand, we prefer to think that meaning in life is a kind of cognition and feeling of the meaning of life, which can be seen as a psychological construct.
Point 3: Similarly, although the Authors describe the meaning of "cognitive failures" there is no explanation provided how they interpret this phenomenon (cognitive impairment? neurodevelopmental deficits? reactive disturbances?)
Response 3: Thank you for your suggestion. We re-clarify the definition of cognitive failures and explain the causes of cognitive failures as follows----”Cognitive failures are cognition based slips and lapses in simple tasks that a person can usually perform without errors, including attention, memory, and motor failures[19]. For example, entering a room but forgetting what to look for, throwing away something that should be kept, etc. Some studies have pointed out that the occurrence of cognitive failures may be related to insufficient cognitive resources[20], negative affects[21], and personal traits such as trait anxiety and neuroticism[22,23].”
Point 4: Although the Authors discuss some mechanisms which may be responsible for association between meaning in life and cognitive failures and mindfulness I would appreciate some more consistent theoretical framework linking these ideas.
Response 4: Thank you for your suggestion.We introduce a theoretical theory of depression in the last paragraph of the introduction to integrate the research variables of this study. The specific modifications and supplements are as follows----”The cognitive vulnerability-stress theory of depression [52] illustrates that cognitive vulnerability and negative events interact with each other to influence the occurrence of depression. Frequent cognitive failures, as negative events, can induce more negative emotions in individuals. Meaning in life and mindfulness can also be considered as individual cognitive vulnerability. Therefore, this study hypothesized that the interaction of cognitive failures, meaning in life, and the level of mindfulness could jointly affect the generation and development of depression.”
Point 5: Method
The Authors mentioned that the vast majority of the participants were not-only children. Forgive my ignorance: taking into consideration the one-child policy in China I wonder how it is possible. Could you explain that to me and other non-Chinese readers?
Response 5: Thank you for your question. I would be delighted to answer your question based on my knowledge. China's one-child policy was enacted and implemented around 1980 in order to control the excessive population pressure in China at that time, so in the 80s and 90s, it was really dominated by one-child families. But around 2000, Chinese couples were allowed to have a second child as long as they applied for a family planning service certificate. The government completely abolished the one-child policy in 2016. Therefore, non-one-child families are still the majority in China now.
Point 6: Psychometric tools: it should be clarified if the questionnaires were validated for children and adolescents age group
Response 6: Thank you for pointing this out. We have supplemented the related information and research about the applicability of the psychometric tools used in this manuscript to the adolescent population.
Point 7: Discussion
In the results description and in the discussion section the Authors mention that "meaning in life affects depression". This and other similar expressions suggest the causative relationship between meaning in life and depression. Similar interpretations are suggested for mindfulness and cognitive failures. Such conclusion definitely cannot be drawn from a cross-sectional study. Cross-sectional studies can detect an association, but not causative effects. Maybe lack of meaning in life is just another depressive symptom, not the cause of depression?
Response 7: Thank you for your careful review. Indeed, as reviewer point out, cross-sectional studies do not draw direct conclusions about cause and effect. What this paper intends to express is the influence that these important factors may have on adolescent depression and the specific relationship among them. According to the suggestions of reviewer, the discussion part of the revised draft is more rigorous in the expression of variable relationship, and the relevant expression of causal inference is deleted.
Point 8: Implications
In the implications section the Authors present opinions which are not supported by the study results (e.g. "Schools and families should guide teenagers to use mobile phones properly to avoid them wasting too much time and energy on mobile phones" - the study is not about mobile phones!) This should be avoided and the section rewritten.
Response 8: Thank you for your valuable comments. We deleted the irrelevant information and rewrote the paragraph as follows----”Secondly, cognitive failures is closely related to insufficient cognitive resources. Therefore, setting a clear life goal and maintaining a positive and optimistic emotional state can reduce the depletion of cognitive resources by negative emotions such as boredom and anxiety, thus reducing the occurrence of cognitive failures and depression”.
We sincerely thank the reviewers for taking the time to review our manuscript and for their valuable suggestions.
Reviewer 2 Report
The approach to the topic is unique, and in addition, the study can be regarded as filling a gap in terms of the examined age group.
Regarding the incidence of adolescent depression, the difference between Chinese and international data is quite large (rows 28-32). The authors fail to reveal or explain the background of the difference (e.g. methodological differences, other factors).
The formulation of the three hypotheses is well conceptualized, although the introduction of the second hypothesis is somewhat imprecise (row 83). The rewiever suggest replacing the word “speculate”.
The presentation of the study's methodology and measuring instruments is informative and accurate. The presentation of the results and the correlations between the variables is thorough. However, it does not make sense to include the mean value of “gender” (as a nominal variable) on Table 1.
Author Response
Point 1: Regarding the incidence of adolescent depression, the difference between Chinese and international data is quite large (rows 28-32). The authors fail to reveal or explain the background of the difference (e.g. methodological differences, other factors).
Response 1: Thank you for your valuable comments. According to the problem you pointed out, we have carefully reviewed and compared relevant literature, and the large difference in the prevalence rate of depression between international and Chinese adolescents mentioned in our manuscript may be caused by the difference in the time range of investigation, the definition and diagnostic methods of depression, and the age range of the subjects in the two studies. Therefore, we re-updated the data based on an international paper published in 2022. From the latest data, there is not much difference between the international data(34%) and the Chinese data(28.4%). Thank you for pointing out this constructive question. We have supplemented relevant data in the revised manuscript.
Point 2: The formulation of the three hypotheses is well conceptualized, although the introduction of the second hypothesis is somewhat imprecise (row 83). The reviewer suggest replacing the word “speculate”.
Response 2: Thanks for your careful suggestion, We replaced the sentence "we can speculate" with "relevant theories and research results implied that". Meanwhile, we have added the concept explanation and influencing factors of cognitive failures in "1.2. Cognitive failures as a mediator" of the revised manuscript, and supplemented the theoretical basis of the current research hypothesis in "1.4. The present study".
Point 3: The presentation of the study's methodology and measuring instruments is informative and accurate. The presentation of the results and the correlations between the variables is thorough. However, it does not make sense to include the mean value of “gender” (as a nominal variable) on Table 1.
Response 3: Thanks for your careful suggestion, we have deleted the row of “gender” in Table 1.
We sincerely thank the reviewers for taking the time to review our manuscript and for their valuable suggestions.
Reviewer 3 Report
Thanks for providing me with this opportunity to read the manuscript. It is an interesting study. The authors addressed an important issue in China. Junior high school students have experienced a heavy workload, and high pressure as they need to face lots of examinations that are essential for them to be accepted to be in a high-ranking senior high school. This might lead to a consequent likelihood to be accepted to be in a high-ranking university in China. In this regard, parents, teachers, and children all face a severe competition and stress. Mindfulness is very important and urgently needed to be researched in the Chinese context.
However, the concerns are mainly on the research tools. They are originally from English. Personally, I am wondering how were they applied in a Chinese cultural context? The previous literature was supported by scholars who validated the tools and then published in Chin. Ment. Health. J.
Personally, I think that the authors need to update the literature to find more international studies that have provided the validation in China. Otherwise, for international audience, they are difficult to understand Chinese.
Author Response
Point 1: However, the concerns are mainly on the research tools. They are originally from English. Personally, I am wondering how were they applied in a Chinese cultural context? The previous literature was supported by scholars who validated the tools and then published in Chin. Ment. Health. J.
Personally, I think that the authors need to update the literature to find more international studies that have provided the validation in China. Otherwise, for international audience, they are difficult to understand Chinese.
Response 1: Thank you for your valuable comments. In view of your suggestions for the research tools, we earnestly make modifications and make the following explanations.
All the scales or questionnaires used in this study were the Chinese version adapted by previous Chinese researchers. They adapted the Chinese version of these scales or questionnaires and tested their validity through the following steps: First, the English version of the scale was translated into Chinese by researchers and English professionals using the "Back-translation" procedure recommended by Brislin(1970). Then, the scale and several other related scales with good reliability and validity were used to test Chinese subjects. Third, the questionnaire items were screened through item analysis, reliability analysis and validity analysis, and finally the Chinese version of the questionnaire was formed.
Meanwhile, we also cite some international studies that also used these Chinese research tools to support the validity of the Chinese version of the questionnaires or scales used in this manuscript (please see the revised references on the measures section of the manuscript for details).
We sincerely thank the reviewers for taking the time to review our manuscript and for their valuable suggestions.